# How to Read the Quran in Religious Islamic Education: What Educators Can Learn from the Work of Mohammed Arkoun

Iddo Felsenthal * and Ayman Agbaria *

Department of Policy and Leadership in Education, Faculty of Education, University of Haifa, Haifa 3498838, Israel
* Correspondence: iddo.felsenthal@gmail.com (I.F.); aagbaria@edu.haifa.ac.il (A.A.)

**Abstract:** The study of the Quran is central to Religious Islamic Education (RIE). Exposed to different political and social pressures, teachers in RIE still struggle between traditional approaches concerning the divine nature of the Qur'an and more secular approaches emphasizing the historicity and critical analysis of the religious text. Mohammed Arkoun (d. 2010), an Algerian-born Sorbonne professor, offered a hermeneutical methodology for reading the Quran that was Western, scientific, and critical, and at the same time related to the Living Islamic Tradition, faith, and thought. The article analyzes Arkoun's methodology for reading the Quran and its possible implications on RIE in a way that creates a space for creativity, criticism, and dialogue between worldviews and opens new possibilities for the faithful to teach and learn the Qur'an.

**Keywords:** Mohammed Arkoun; religious Islamic education; reading the Quran

## 1. Introduction

The study of the Quran and its interpretations (henceforth, reading the Quran) is foundational to Religious Islamic Education (RIE). Indeed, the Quran is widely considered the most important source of Islamic jurisprudence, the main carrier of the Islamic credo, and the primary text upon which Muslims draw to practice their religion, cultivate their Muslim character and identity, and form their communities of believers (Berglund and Gent 2019). Berglund and Gent (2018) noted that:

> "[A]t the heart of knowledge-seeking is learning and studying the revealed knowledge as set out in the Qur'an, believed by Muslims to constitute the final message of Allah to humankind as revealed to the Prophet Muhammad over the course of the last 23 years of his life and passed on by him through recitation" (p. 127)

As the Quran is at the heart of knowledge-seeking, the methodology of reading the Quran becomes imperative for RIE. How to approach Quranic texts and to what extent one can engage critically with classical interpretation dictate the possibilities and impossibilities of the religious imagery and experience of the teachers, let alone their pupils. In this regard, the medium is the message, as the methodology of reading the Quran instates the reader's assumptions and presuppositions about the text's sacredness, relevance, and significance. Furthermore, the methodology not only reflects the readers' presuppositions but, to some extent, constitutes the positionality from which the text is approached for both purposes of worship and finding answers, be they existential or instrumental. In a word, the methodology molds, to a lesser extent, the answers the Quran provides for Muslim believers and, to a greater extent, the questions the believer can ask in search of meaning and significance.

There is, of course, more than one way to read the Quran. Akbar juxtaposes "a number of innovative hermeneutical approaches, often situated under the umbrella of contextualization" to "the exegetical discourses of literalism/textualism" (Akbar 2017, p. 1). The

latter supports the idea "that the literal meaning of the text is privileged over other forms of meaning in the process of interpretation" (Akbar 2019, p. 1). The contextual reading relates to reformist approaches in Islam and to the call for "reinterpretation of the Qur'an and hadith and re-opening the gate of ijtihad" (Elmi 2014, p. 278). The literalist reading relates to the Salafi approach but can also be found among the Usuli Twelver Shi'ites and "has become one element in a more general movement of textual devotion" (Gleave 2012, p. 176). Sadaalah (2004) adds to these two approaches and lists four orientations of contemporary Islam: secularist, traditionalist, modernist/liberal, and fundamentalist/Islamist. These orientations generate different approaches to education and reading the Quran. To these, one can also add the Sufi or mystical approach to reading the Quran (Calis 2020).

It seems safe to argue that the different approaches to reading the Quran revolve around two issues: the historicity of the Quran and its divine nature, and the authority of the classical exegeses. Namely, is the Quran a credible historical document that tells of true events? In addition, to what extent reading the Quran might engage critically with the Tafsirs, Muslim exegeses and commentaries (e.g., al-Tabari (839–923), al-Zamakhshari (1075–1144), al-Razi (1149–1209), and Ibn Kathir (1301–1373).

In the Orthodox, canonical, if you will, readings (more about Orthodoxy to come), for example, the historicity of the Quranic tales of prophets is undeniable, making it almost impossible for teachers in RIE to question the choices and decisions of the Quranic narratives, which are perceived as literally and absolutely real occurrences and true experiences. Thus, teachers of RIE who comply with the authority of the exegeses in this vein of orthodoxy have little room and legitimacy, if you will, to deliberate on missing information in these tales, elicit insights regarding mistakes and misjudgments of the prophets, compare these tales to the Jewish and Christian biblical narratives, express critical stances, and be engaged in reflective discussions.

Another substantial pressure on RIE teachers comes from modern ideological interpretations, mainly the nation-state and political Islam movements. Doumato and Starrett (2007) review numerous case studies, including Egypt, Syria, Jordan, Turkey, and Saudi Arabia, in which governments have fashioned generic and unified versions of Islam within the framework of the school curriculum that confer and stipulate the stability of the regime. Whether religious or secular, each state has tailored its own official unified version of Islam that would grant it legitimacy, promote civic ideals of discipline and conformity, and serve its political interests. Another version of unified Islam that pressures and affects the way Quran is read, interpreted, and taught in RIE is the version of political Islamic movements. Ideologized exegeses have granted a place of (dis)honor for works such as Fi Zilal al-Quran by Sayyid Qutub or Tafhim al-Quran by Abu A'la al-Mawdudi in many of the works pertaining to the questions of contemporary interpretations of the Quran.

Against this backdrop of different reading approaches and ideological pressures on RIE teachers, we argue that the methodology of Mohammed Arkoun in reading the Quran suggests new directions and reflective spaces for creativity, criticism, and dialogue for reading the Quran in RIE. Employing a hermeneutical approach, Arkoun wishes to consider "the various and irreducibly different Muslim traditions that have developed in diverse cultural and ethnic environments" as what "constitutes Islam" (Harrison 2010, p. 218). His work, therefore, goes "beyond hegemonic intellectual binaries of modernity versus tradition" (Kersten 2011, p. 25). "He suggests daring methods and concepts", criticizing both Western and Muslim thought and thinkers (Coury 2018, p. 171). On the one hand, Arkoun's methodology could enable educators in RIE to read the text in a critical way using Western academic and scientific tools; on the other hand, the same methodology could enable the same educators to include expressions of faith and religious tradition in their reading of the Quran. This combination set within a hermeneutical methodology opens new possibilities for the faithful to teach and learn the Quran as an integral part of RIE and confessional education.

*A Few Words about Mohammed Arkoun*

Mohammed Arkoun embodies through his biography and work the tension between the state and religion, and liberal democracies and Muslim communities in the "West". Born to a Berber family in Algeria in 1928, Arkoun taught at the Sorbonne in Paris until his death in 2010. He could be associated with a group of "new Muslim intellectuals" who:

> "[E]xhibit a critical but constructive view of the Islamic tradition. Their comprehensive and inclusivist ways of engagement draw on an intimate familiarity with the Islamic heritage, or *turath*, and an equally solid knowledge of recent advances in the humanities and social sciences developed within the Western academe" (Kersten 2011, pp. 23–24).

While calling for analytical sociological, anthropological, hermeneutical, and post-structuralist methods to be implemented in research on Muslim thought and tradition, Arkoun conducts his discourse from within both the academic Western world of thought and Muslim Tradition and thought (Ahmad and Lakhvi 2019; Harrison 2010; Kersten 2015; Shaikh 2004).

While, as mentioned, Arkoun was influenced by postmodern thinkers, he refused to identify as one. For to him this mode of thought belonged to the Western space of thought. He, rather, defined his work as "meta-modern logic" (Kersten 2011, p. 40). Writing on Akroun has emphasized the way he was influenced by Western and particularly French Postmodern thought, to the point that his "assessment of modernity and Islam is not a general critique of religion based on traditional religious arguments but rather on postmodern critical theory" (Ahmad and Lakhvi 2019, p. 16).

It is important, however, to point to Muslim works that Arkoun himself found influential on his thinking. Arkoun's focus on Miskawayhi, al-Tawhidi, and Ibn Rushd highlights to what extent Arkoun's humanistic and rationalistic approach draws on, if not resides within, Muslim philosophical and religious thought. In Arkoun's words:

> "The mindset, cognitive options and activities of these intellectuals (Miskawayh and Tawhidi—I.F. and A.A.) in the Islamic polity (almadina) can be compared to those displayed on a larger scale, with more appropriate scientific tools and social support, by European philosophers and writers of the seventeenth and eighteenth century" (Arkoun 2012, p. 420).

Without diving into a detailed account and notwithstanding other influences, Arkoun's work was perhaps most influenced by the poststructuralist hermeneutics of Paul Ricoeur that enabled him to "synthesize these various strands of thought into a new way of engaging in Islamic studies" (Kersten 2011, p. 25).

As a theory or philosophy, hermeneutics is defined as the interpretation of meaning (Butler 1998). This kind of interpretation of meaning focuses on how "the meaning of religious language is generated through the creative interaction of readers with texts" (Harrison 2010, p. 208). Here, interpretation involves entering into the interpretive norms of a community, in which meaning is found and operates in the historical context of the interpreter and the interpreted (Butler 1998; Coyne 1995). The critical perspective, as seen in the work of Habermas (1972), uses interpretation to challenge conventional wisdom in a community to address power asymmetries and achieve emancipatory goals (Butler 1998).

According to Arkoun, all believers are locked in what he, following Paul Ricoeur, calls a "hermeneutic circle" in which "everyone must believe in order to understand and understand in order to believe" (Arkoun 2016b, p. 122. All translations from French into English were made by the authors). He, therefore, aims his hermeneutics to deal with "the relevance peculiar to the discourses derived from the interpretation of a written text" (Arkoun 2012, pp. 137–38). It is here, in the tension between interpreting the norms of a community and challenging and critically thinking about these norms, that the work of Arkoun offers an opening of space for thought. He refers to his "hermeneutic quest" as an "exit thought" (sortie pensée), and through it he wishes to "clarify the ambiguous meaning of religion", which, on the one hand, is promoting the dignity of the human person, and,

on the other, is "manipulated by social actors who subordinate the actualization of this promotion to desires of power" (Arkoun 2016b, p. 390).

Smith (1991) locates the aim of hermeneutics as striving for "human freedom" while "problematizing the hegemony of dominant culture in order to engage it transformatively" (p. 195). In this sense, Arkoun's hermeneutics is exactly what Smith describes as "a link between social trouble and the need for interpretation" (p. 188).

Therefore, Arkoun's work offers a way out for RIE teachers because his hermeneutic efforts wish to both understand the meaning of the Quran as constructed by Muslim communities throughout the years as well as think of these traditions and understandings critically and try to transform them from within. This is related to and intensified by Arkoun's own positionality within the two systems of thought of Western academic-scientific Tradition and Muslim Traditions.

In this paper, we will show how Arkoun's hermeneutic reading of the Quran might influence and offer solutions for teaching the Quran in RIE. Arkoun analyzes four faces or layers in that reading: the Word of God, the Prophetic Discourse, the Official Closed Corpus, and the Living Tradition. While the first two layers, the Word of God and the Prophetic Discourse, are, to Arkoun, the *Quranic Fact* and are open to (almost) endless options of interpretation as they are part of the Revelation, he terms, The Official Closed Corpus and the Living Tradition—the *Islamic Fact*, a human interpretation of the Revelation that has been closed through writing and "normative devices", which can be related to the process of orthodoxization (Arkoun 2012, pp. 308–9).

We shall follow his methodology of reading through the layers, analyze it, and show an example of how he uses his methodology on the story of Yusuf. Then, we will analyze his analysis of the story of Yusuf and show how this methodology could affect the teaching of the Quran in RIE by using both critical tools drawing on Western academic Tradition and giving room and respect for Muslim Tradition, thought, and faith.

## 2. Four Faces of the Quran

### 2.1. The Word of God

The first layer in Mohammed Arkoun's analysis of the Quran is the Word of God. Relying on verse 27 from Surat Luqman (31): "If all the trees on earth were pens and the ocean was ink, refilled by seven other oceans, the Words of Allah would not be exhausted", Arkoun finds the Word of God to be "real and infinite" (Arkoun 2016b, p. 15). This is similar to the way John Hick described religious realism as "the view that the objects of religious belief exist independently of what we take to be our human experience of them" (Hick 2004, p. 172).

This ultimate Reality, the divine, according to Hick, "exceeds human thought and language" (Agbaria 2022, p. 10). For Arkoun, this Reality of the Word of God transcends "all the linguistic performances realized or realizable in human languages, therefore all the human uses of thought" (Arkoun 2016b, p. 106). It is because of this that the Word of God, according to Arkoun, cannot be subject to critical scientific tools and it "awaits to be restored in the fullness of its intention as a challenge to all men . . . " (Arkoun 2016b, p. 65).

As transcendent as it may be, the Word of God is the basis of the religious experience. It provides the believers, throughout the Quran with "clear, eternal indisputable norms, teachings and ideal commandments to enlighten this life and lead to Salvation" (Arkoun 2012, p. 58). Making *Shari'a* into "Divine Law", it functions "as the source and fundamental root of every type of knowledge" (Arkoun 2012, p. 59), and "serves as the ultimate and inevitable point of reference for every act, every form of behavior and every thought of the faithful" (Arkoun 2012, p. 79). To Arkoun, the whole Muslim ethos is "nurtured" by the Word of God in a "spontaneous, emotional integration of the sacred world into everyday life by believers . . . " (Arkoun 2012, p. 312).

Furthermore, the Revelation "feeds a living tradition that permits the community to resupply itself with the radical novelty of the original message" (Arkoun 1994, p. 34). This

means that the Word of God not only influences and dictates the traditions, but it is also the source for the renewal of the religious experience.

The Word of God is the basis for a necessary comparative study and analysis of religions, as it exists in the Bible, the Scriptures, and the Quran (Arkoun 2016b, p. 15). Relating to the term *ahl al-kitab*, which appears 31 times in 11 different *suras* (Hoffman 2018), Arkoun names the monotheistic societies—*the societies of the Book-book*, "where the Holy Book continues to shape and direct the production of books ... " (Arkoun 1994, p. 33). The Book in these societies creates a transposition, changing the "collective destinies" under the direction of the prophets into a "book" that is a living tradition (Arkoun 2016b, p. 319). This is the basis for "a comparative theology of revelation" (Arkoun 1994, p. 31). This comparative theology of revelation can also be the basis for comparative methodologies in RIE, where educators could and should refer to other religions in teaching the Quran.

### 2.2. The Prophetic Discourse

The second layer is the Prophetic Discourse, where a prophet, in the case of Islam, Muhammad, recites the Word of God, the Revelation, to the believers. Arkoun, after translating and quoting verses 1–5 of *Surat az-Zukhruf* (43), claims the recitation of the Prophet Muhammad is the "Arabic Quran", which is differentiated from *Umm al-Kitab*, kept in the presence of God, and as we have seen, as the Word of God, which is the absolute truth, unattainable by human knowledge and perception. Daniel Madigan claims it is "the point where the timeless authority and insight address the time-bound human condition" (Madigan 2001, p. 77). The Prophetic Discourse is "mediating the Word of God" (Arkoun 2012, p. 346), so that humans would "understand the substance of the Message" (Arkoun 1994, p. 31). In this sense, Arkoun's analysis of the Prophetic Discourse is "humanistic", as defined by Ali Akbar:

> "I use the term 'humanistic' to refer to any approach to interpreting the Quran which rests on the view that revelation is not only dependent upon its initiator (God) but also its recipient (Muhammad), in the sense that the latter is more than a mere passive recipient of the revelation. From a humanistic perspective, revelation does not involve a one-sided process in which God communicates to His prophet without the latter contributing anything to the content of revelation. That is, the content of revelation should not only be ascribed to God's authorship or influence, but also, in some important aspects, to its human recipient as well." (Akbar 2019, p. 3).

Arkoun uses the term *discourse* because the Prophetic Discourse is recited, "heard, not read" (Arkoun 1994, p. 30). The Quran, he reminds us, "before it became a graphically fixed text was a speech, a word (parole)", and liturgically it remains so to our days, recited by Muslims around the world (Arkoun 2016a, p. 11). This also allows for further inquiries into the context, history, society, and culture of the time, referring to layers of the Quran as discourse. As Madigan points out, the Quran relates to itself "more in terms of process than fixed content" (Madigan 2001, p. 144).

To Arkoun, this discourse is more linguistic than theological (Arkoun 2016b), and he analyzes it, accordingly, relying on the renowned linguist Émile Benveniste (died 1976): "The totality of the Quranic discourse reveals three protagonists: a speaker-author (*qa'il*), an addressee-enunciator (Muhammad) and a collective recipient (the people)" (Arkoun 2012, p. 69; 2016a, p. 15). The collective recipients are "equal and free" (Arkoun 2012, p. 69). They share an access to the same enunciation and share a position in the discourse. The recipients are free "because they respond immediately by assent, understanding, rejection, refutation or the demand for further explanation" (Arkoun 2012, p. 69). This was true for the people at the time of the Prophet Muhammad, but it is also true for people who read and hear the Quran today.

The Prophetic Discourse, according to Arkoun, accompanies the Founding Experience of Islam, which relates to the period of twenty years of Muhammad's prophetic activity (612–632). In this period, the Quranic discourse presented was "gradually received as the

authentic Word of God" (Arkoun 2016b, p. 277). It was also accompanied by "the speech of Muhammad, living and acting among his own" (Arkoun 2016b, p. 277), which would later become the *Hadith*. Arkoun stresses, however, that those two discourses, represented today in the *Mushaf* (Arkoun refers to it as Official Closed Corpus, see below) and the literature of the *Hadith*, "raise difficulties that modern historical criticism has not yet succeeded in overcoming" (Arkoun 2016b, p. 277).

Arkoun does, however, suggest an analysis of the "prophetic function", as he calls it, according to the Quran and the life of Muhammad. This function lies in the balanced duality between historical action and religious experience. Each action of the prophet "opens a hope for men", as Muhammad's actions are not only viewed as events in his space and time but also through the generations as a religious experience. To Arkoun, "[T]his is the meaning of Revelation which 'descends' whenever the Prophet commits, by a decision, the spiritual future of the community through a contingent situation" (Arkoun 2016a, p. 19). In other words, the actions of the Prophet Muhammad, informed by the Revelation of the Word of God, dictate and influence the religious experience of generations to come. It is "this power of Revelation" that makes such actions meaningful to groups of believers, those who surrounded Muhammad and those who came later (Arkoun 2016b, p. 395). This is, again, not unique to Islam or to Muhammad. Arkoun lists different Founding Experiences, such as the Israelite Exodus from Egypt and the Passion of Christ.

According to Arkoun, the Prophetic Discourse contains, therefore, three vectors of interpretation that can be found in every Quranic verse: (1) **Historicity of invented values**—that is, the historical, cultural, social, and political conditions that shaped the way different agents formed values and norms at the "Founding Experience". (2) **Axiological activity**—that is, the creation of values and norms that permeate through time and are interpreted as Living Tradition in our days. (3) **Metaphorical organization of expression**—that is, the metaphoric potential of each verse that stems from the Marvelous in the Quran and its Divinity. These vectors intersect in each Quranic enunciation, and to separate them "by treating the text exclusively as a historical, linguistic, or legal, or ethical document" will be distorting the "function of meaning" (Arkoun 2016b, p. 320).

The Prophetic Discourse is full of meaning. Its "symbolic capital" has "the power of promoting creative thought" (Arkoun 2012, p. 347). In order to access this capital and to (re)interpret the Prophetic Discourse we need to use both Western, modern "historical criticism" and the Islamic orthodox paradigm of thought as we cannot change a "language of mythical structure to a simple denotative language" (Arkoun 2016a, p. 21). In order to use them, we must deconstruct them both first (Arkoun 2012, pp. 264–65).

Arkoun, therefore, suggests analyzing and deconstructing each vector of meaning (history, axiology, and metaphor) and combining them to regain the function of meaning. He also suggests deconstructing them and using both the Islamic-orthodox-Traditional and modern-scientific-critical cognitive frameworks to do so.

In RIE, as we shall see, this can be useful for educators in analyzing and finding relevant meanings in the Quran, with their students.

### 2.3. Official Closed Corpus

The third layer of the reading, the Quran is what Arkoun terms the Official Closed Corpus (OCC). At this level, the Quran, which up to this point had been a verbal discourse, is written down and becomes *Mushaf* (Arkoun 2012, p. 86). Arkoun calls it *official* because it was created by "authorities" of the community and *closed* "because nobody was permitted any longer to add or subtract a word, to modify a reading in the Corpus now declared authentic" (Arkoun 1994, p. 33).

This transition from a verbal discourse to a written text is a "decisive, irreversible, historic event" (Arkoun 1994, p. 33) and a break from the previous layers. Like in the Word of God and Prophetic Discourse, Arkoun points out that this happened in Judaism and Christianity as well (Arkoun 1994, 2012).

To Arkoun, the opposition between verbal discourse or *orature* to a written book or *écriture*, is part of a larger opposition between "(1) the centralizing state, (2) *écriture*, (3) the learned élites and (4) orthodoxy" and "(1) segmentary societies, (2) *orature*, (3) culture which is called popular . . . and (4) heterodoxies" (Arkoun 2012, p. 70). This kind of opposition is typical of the way anthropologists view diversity in Muslim tradition, in which orthodoxy becomes "merely one (albeit invariable) form of Islam among many" (Asad 2009, p. 8). Seen this way, Orthodoxy becomes "a relationship of power to truth" (Asad 2009, p. 14), "claiming its authority from sacred texts rather than sacred persons" (Asad 2009, p. 8).

Arkoun emphasizes the political element in the establishment of the OCC and orthodoxy. The creation of the OCC is closely related to the establishment of the Imperial Muslim state. It is through the establishment of the Muslim empires, termed "the imperial moment" (le moment imperial) by Arkoun, that certain interpretations of the *mushaf* were accepted and others were rejected and marginalized. This holds a significant legitimizing role for the regimes from the days of the first Caliphs (aptly referred to in Muslim Sunni tradition as the Righteous Caliphs, literally *those who walk on the straight path* or in Greek—Orthodox) through the days of the Umayyad and Abbasid dynasties and all the way to modern-day authoritarian regimes in Muslim countries (Arkoun 2016b, p. 322).

The political legitimization that stems from the creation of the OCC and orthodoxy— brings forth the important authority nexus of Ulama and regimes (Arkoun 2012, p. 275). As Arkoun, like other anthropologists, sees orthodoxy as one form of Islam out of many, he thus reopens the issue of authority, relativizes it, and allows room for humanistic autonomy.

Whatever the repercussions of the OCC on orthodoxy are, the *mushaf* is the written text we have before us in reading the Quran. If the Prophetic Discourse is the layer upon which Arkoun focuses his (re)interpretative efforts, combining Living Tradition and academic scientific tools, it is the OCC that provides Arkoun with the text for analysis and (re)interpretation. We are left only with the OCC, as "the strict linguistic relation between the Official Closed Corpus and the Quranic Discourse during the situations of enunciation of each verse is lost forever" (Arkoun 2016b, p. 319).

In RIE, the OCC is not only the text read; rather, educators should analyze with their students the way the *Mushaf* came to be referring to the social, political, and cultural realities of Muslim history.

### 2.4. Living Tradition

The fourth layer in Arkoun's suggestion for reading the Quran is the Living Tradition. The Living Tradition, according to Arkoun, contains the Quran (as Official Closed Corpus), the *Hadith* (or prophetic traditions with the addition of the teachings of the Imams, in the case of the Shi'ites), and the *Shari'a* ("the legal codification of God's commandments") (Arkoun 2012, pp. 301–2).

Living Tradition derives from the OCC. It is the book that comes after the Book, or the "secondary corpora" created by Islam, Christianity, and Judaism (Arkoun 2012, p. 82).

This split of existential realities, or the one text to numerous texts, is "the social, cultural and political construct of the "living Islamic Tradition'" (Arkoun 2012, p. 255). In this sense, Living Tradition is diverse not only because society, culture and politics are diverse within the Muslim world; rather, its diversity derives from its structure, from being an earthly interpretation of the divine that has become text.

Arkoun's view of a diverse Living Tradition (we will deal with the capital "T" soon) is part of a change in the way anthropology and philosophy think about Islamic Tradition. This rethinking, relying on Alasdair MacIntyre, "have called into question the modern prejudice that tradition must always be in ontological opposition to rationality and negotiation" (Anjum 2007, p. 661). Tradition becomes a Discursive Tradition (Asad 2009; Sulaiman 2018). This does not only mean a diversity of traditions, but it relates to "a past (when the authentic practice was instituted) and a future (how a correct performance and its fruits can be secured in future) through a present (how it is linked to other practices, institutions, and social conditions)" (Anjum 2007, p. 661). In other words, Islamic tradition, to Ovamir

Anjum is "not determinative but interpretive" (Anjum 2007, p. 667). To Mohammed Sulaiman this is "the distinctive feature" which allows for its diversity and "multiplicity of Islamic *local* orthodoxies" (Sulaiman 2018, p. 143, italics in the original text).

Arkoun, however, sees the Living Tradition as less diverse. He points out its origin in text and reminds us that "where there is text, there is selection, that is elimination of methods, schemes and concepts conforming to the requirement of each established school" (Arkoun 2016a, p. 40). This had created "a homogeneous space of representation, projection and expression" (Arkoun 2016b, p. 164).

This forms a "**dogmatic enclosure**" (Arkoun 2012, p. 94, emphasis in the original text) exercising "control over the **thinkable** and the **unthinkable**" (Arkoun 2012, pp. 30–31, emphasis in the original text).

The terms *unthought* and *unthinkable* in the work of Mohammed Arkoun are extremely important. It means the "accumulated issues declared unthinkable in a given logosphere" or:

> "[T]he linguistic mental space shared by all those who use the same language . . . to articulate their thoughts, their representations, their collective memory, and their knowledge according to the fundamental principles and values claimed as a unifying *weltanschauung*" (Arkoun 2012, p. 17, italics in the original text).

"The obligation to rethink and rewrite" the Islamic thought's "entire history within the dialectic framework of the **thinkable/unthinkable, thought/unthought**" is one of Arkoun's most emphasized contentions (Arkoun 2012, p. 19, emphasis in the original text). The Living Islamic Tradition, therefore, shapes *thought* and *unthought* of the Islamic logosphere.

Furthermore, it is, after all, Tradition with a capital "T", seeking to "impose its supremacy, disregarding and, where possible, eradicating all previous local traditions" (Arkoun 2012, p. 302). The dialectics of *thought/unthought* run along similar lines as the dialectics of written/oral culture, "with writing, learned elites and orthodoxy (religious and political) on one side and fragmented societies (tribes, clans, and patriarchal families), oral 'dialects' and cultures, 'heterodoxies' . . . on the other" (Arkoun 2012, p. 148). To Arkoun, each community preserves "original orthodox teachings, excluding the others as heretical sects" through an "arbitrary selection of traditions referred to the Prophet, the Companions and the Imams" (Arkoun 2012, p. 295). As a result, religion "serves social actors" who "assure their own control over symbolic goods" (Arkoun 1994, p. 47). Thus, "none escapes this dialectic of the powers and the residues" (Arkoun 2012, p. 31).

In this dialectic historical process, "voices are silenced, creative talents are neglected, marginalized or obliged to reproduce orthodox frameworks of expression . . . " (Arkoun 2012, p. 16). However, claims Arkoun, the residues persist and "[T]he Islamic orthodox paradigm has never exercised its full impact on all groups and sectors" (Arkoun 2012, p. 265).

As "meanings, effects of meaning and horizons of meaning do not emerge only where hegemonic reason is active", Arkoun wishes to "hear voices reduced to silence, heterodox voices" and bring them back to the interpretation table (Arkoun 2012, p. 32). This should be done by "[A]n ethno-sociological survey . . . of this neglected but exuberant Islam" (Arkoun 2012, p. 154). More than creating diversity, this deconstruction of Tradition, serves to differentiate between orthodoxy as a "militant ideological endeavor, a tool of legitimation for the state and the 'values' enforced by this state" (Arkoun 2003, p. 22) or in Asadian terms, "a relationship of power to truth"; and between what Sulaiman, following Asad, calls "a universal Orthodoxy" (Sulaiman 2018, p. 143), or in Arkounian terms, "religion as a way proposed to man to discover the Absolute" (Arkoun 2003, p. 22).

The input of Living Tradition in teaching the Quran in RIE is crucial to both understanding the diversity of said tradition and its relevance to students' and teachers' identities.

### 3. Arkounian Methodology of Reading the Quran

The hermeneutical methodology of Mohammed Arkoun involves three elements: Critical thinking of the *unthought*, going "back and forth" between details and the whole, and a fusion of horizons through theo-anthropology.

### 3.1. Critical Thinking of the Unthought

To Arkoun, being critical means subjecting all texts to a critical deconstruction whose focus is the *unthought* while "pushing the boundaries" and opening "new avenues of thought" (Arkoun 2012, pp. 4–5). His "systematic deconstruction of the original texts" is combined with "critical review of modern studies" (Arkoun 2012, p. 13).

The critical analysis should be "out of active resistance" focusing on the *unthought*. This is a "**subversion** of reason for the sake of reason" (Arkoun 2012, p. 8, emphasis in the original text) or a "radical critique of reason" (Arkoun 2012, p. 5). When Arkoun writes **subversion** (and emphasizes it), he means to "open wide for the first time in the intellectual history of Islamic thought, all the spaces for scientific research and free critical thought" (Arkoun 2006, p. 159). This subversion "takes charge of the more obscured or arbitrarily sacred subjects, the most protected traditions by religious and/or political taboos" (Arkoun 2006, p. 163). Furthermore, "reflexive hermeneutics" is itself "subversive" (Arkoun 2016b, p. 390).

But this critical thinking of the *unthought* is not turned only to Islamic Tradition or thought; Arkoun's methodology demands that this critical tool be directed at Western thought as well. It needs to target "modern thought with its mythoideological excesses" (Arkoun 2006, p. 162). Western thought guided by "a positivist *instrumental* reason" or the "tele-techno-scientific reason" (Arkoun 2012, p. 26, italics in original) should also be subject to critical deconstruction. He wishes to submit "both religious and tele-techno-scientific projects of globalization" to "an encompassing critical reassessment" (Arkoun 2012, p. 281).

Harrison (2010) claims that the basis of "the relevance of hermeneutics to our understanding of religious language" is "[t]he idea that the notion of meaning can be separated from the concepts of truth and falsity" (Harrison 2010, p. 208). In other words, "secular" hermeneutics into religious language can exist because the meaning of that language is not dependent on the truth value of such meaning. The existence of God or the factual occurrence of an event described in the Bible, the Scriptures, or the Quran is irrelevant to hermeneutical questions because they carry meaning whether they are true or not. To Arkoun, however, "[t]he search for ultimate meaning depends on the radical question concerning the relevance and existence of an ultimate meaning. We have no right to reject the possibility of its existence" (Arkoun 2003, p. 25). The "ultimate meaning" is out there, according to Arkounian hermeneutics. Furthermore, to him "true responsibility of the critical reason" should be to better understand "the relationship between meaning and reality" (Arkoun 2003, p. 25). His calls for "subjecting the Quran" to "critical examination" are part of a call for the reader to "decipher" the Quran, postulating implicitly a "true" meaning to the Quran (Arkoun 2016b, p. 67). Thus, the critical thinking of the *unthought* is limited by the search for ultimate meaning and its postulated existence.

Furthermore, one could also contend that when Arkoun is aiming his critical subversion at what Muslims "have long thought as **revelation**" (Arkoun 2012, p. 4, emphasis in original), he merely moves the glass ceiling, the lines of *unthought*, but does not break them, as the Word of God remains in Arkounian hermeneutical methodology, untouched by human reason or examination.

### 3.2. "Back and Forth": Details and Whole

"We must understand the whole in terms of the detail and the detail in terms of the whole" (Agbaria 2022, p. 11), asserted Hans-Georg Gadamer. Dilthey and Rickman (1976) explained, "The whole of the work must be understood from individual words and their combinations, and yet the full comprehension of the details presupposes the understanding of the whole" (p. 115). Following this hermeneutical principle, Mohammed

Arkoun repeatedly asserts the need for a "back and forth" movement between seeing and analyzing the smallest of details and examining the whole picture. We have seen this in the way he constructs the Prophetic Discourse as a Founding Experience. Arkoun asks the reader to both deconstruct each verse through each vector of meaning (historicity, axiology, and metaphor) and look at the verse as a whole (Arkoun 2016b). This is true for all religious texts, as Arkoun urges the reader to "consider the text in its totality as a whole system of inner relations" (Arkoun 2016b, p. 69).

The movement "back and forth" is also essential because it is the "totality" that gives the analyzed text and religious experience their irreducibility. "It is the totality of the text", writes Arkoun in one of his more poetic sentences, "that transports the spirit beyond the space of regular time, and offers it an ensemble of *wonders* (*mirabilia*): the realities that no eye has seen, no ear has heard, but that are eminently worth seeing, hearing and internalizing." (Arkoun 2016b, p. 220).

In order to take into account this irreducibility, this marvelousness, Arkoun suggests the methodology of going "back and forth" between the deconstruction of details, vectors, and aspects of the Quran and the religious experience and understanding of the text and the experience as a whole. Thus, Arkoun helps us to "avoid the methodological pitfall by maintaining a constant toeing and froing" (Arkoun 2012, p. 137).

*3.3. Fusion of Horizons through Theo-Anthropology*

Horizon is a key concept in hermeneutics that represents the limits of one's perspective and viewpoint. This affects how one sees and thinks about the world and is "a standpoint that limits the possibility of vision" and can be described as "the range of vision that includes everything that can be seen from a particular vantage point" (Gadamer 1976, p. 117).

One of the main goals of understanding, in a hermeneutic sense, is to achieve horizon harmony or fusion of horizons. Achieving a horizon of harmony does not simply mean combining perspectives. According to Ricoeur and Thompson (1981), which greatly influenced Arkoun, the fusion of horizons is premised upon the rejection of both objectivism and absolute knowledge. In objectivism, the objectification of the other results on the forgetting of oneself, and in absolute knowledge, universal history can be articulated within a single horizon. Ricoeur claimed that no horizon is closed since it is possible to place oneself in another's point of view (p. 75).

In this sense, Arkoun manifests in himself a fusion of horizons. On the one hand, he holds the horizons of Muslim Living Tradition, thought, religious experience and faith in the Marvelous and the Divine. This "religious marvelous asserts itself" as part of the Quranic text (Arkoun 2016b, p. 216). On the other hand, as we have seen, Arkoun holds the horizon of scientific reason, academic theory, and criticism.

A fusion of horizons, however, is not simply the combining of perspectives. Arkoun suggests, therefore, a field of theo-anthropology of the Revelation. In this field, "all apologetic positions of Islam that resist secularization" will be eliminated, but also "the pretentions of the secular thought to represent a decisive stage of emancipation of reason from imagined beliefs" will be rejected (Arkoun 2016b, p. 199). Then, there will be a "reinsertion of the Revelation" and "reason will invest in new explorations of meaning" (Arkoun 2016b, p. 199). In other words, while exposing Islamic religious thought to the deconstruction of secularization, Arkoun also strives to bring the Marvelous and the religious back into the game of interpretation, influencing reason. It is an intricate, complicated pas de deux.

Arkoun also brings back the whole Living Islamic Tradition. To him, it is "necessary to go back all the way that is traced, marked and imposed by tradition", from the Word of God through the Prophetic Discourse to the Official Closed Corpus to the "interpreted corpora", or as he calls it, "retracing backwards" (reparcourir à rebours) (Arkoun 2016b, p. 318).

The goal of this "retracing backwards" is to look back "with the modern archaeological cognitive project" and thus uncovering "axiological hidden discourse" in all relevant texts of Islam as a Living Tradition (Arkoun 2012, p. 247). Thus, we can follow the interaction between "interpreted Corpus and the social-historical realities" (Arkoun 2016b, pp. 318–19).

In this fusion of horizons, Arkoun wishes to remind us of the "primary function of revelation: to reveal meanings without reducing mystery"; to him, it is the connecting of Western, scientific, deconstructive reason to a relation with "a power equipped with an infinite capacity to signify things, including the truth of being." (Arkoun 1994, p. 42). Arkoun is bringing God and Tradition back to the field of critically interpreting the Quran. He fuses the religious Muslim horizon with the deconstructive, critical, academic horizon. In doing so, he creates what he believes is a "meta-modern horizon of meaning, knowledge and action" (Arkoun 2012, p. 60). New horizons are opening.

### 4. An Example—Surat Yusuf

Arkoun is described as an "intellectually sophisticated religious thinker" (Harrison 2010, p. 217), but in between his convoluted sentences and idiosyncratic terms, there are a few times in his work where he analyzes Quranic verses and *suras*. One such happy occasion is *Surat Yusuf*, the 12th *sura* in the Quran. Arkoun analyzes this *sura* as an example of the way we should (re)interpret the Prophetic Discourse. Analyzing Arkoun's reading of the *sura* will provide us with an example of his methodology, which would be the basis for extracting the educational implications of Arkoun's theories.

Arkoun starts his analysis of the *sura* with verse 3, focusing on the word *ghafil*, which describes the status of Muhammad, "and, therefore, all men" before the revelation (Arkoun 2016b, pp. 320–22). He translates it as *insouciant* in French (which literally translates into English as "carefree"). To Arkoun, this marks a clear ontological break in the world and in people's lives before and after divine revelation. This continues to exist, in the *sura* and in general, as a struggle between humanity before and humanity after Revelation. This underlines the axiological vector that passes like a thread in the story of Yusuf as a rapture between brothers, between man and woman, between the Egyptian culture, law, and religion and the ancient Hebrew monotheistic culture, law, and religion.

Continuing with the axiological vector, Arkoun states that Yusuf is cast away in a strange society without a support group. Going back to a linguistic analysis, Arkoun focuses for a sentence on "*abawayhi*", which appears twice in the *sura* (verses 98 and 99), and which Arkoun analyzes as a "metonym that designates culture and identity of Me against the Other".

Arkoun moves to analyze and deconstruct Yusuf, the hero of the *sura*, claiming that four realities are imposed on Yusuf as a free person. The first reality of a "free person" is a "living alliance in total reciprocity of perspectives with a living God". Here, Arkoun refers to the phrase *la hukma illa lillah*, which appears twice in the *sura* (verses 39 and 66), reminding the reader explicitly of the Kharijite movement in early Islam, which used this phrase as their banner.

The second reality of a "free person" is *tawakkul* a placing of "all his/her confidence-hope in God". Arkoun draws an axiological conclusion from the story of Yusuf that this "does not mean a lazy abandonment of yourself, but a confident, enlightened activity oriented towards the incarnation of the teachings of God in the everyday life of men".

The third reality of the "free person" is that "the alliance God-man, man-God is set as a living solidarity between two creative freedoms". Arkoun explains: "each initiative of Joseph is presented as a personal choice, but the ensemble of choices is registered as a part of the weaving of destiny lead by a superior Will, unfathomable and emancipatory."

The fourth reality of the "free person" is that the actions taken in this "confidence-hope" in God encounter different obstacles and opposition. This, to Arkoun, engenders a new liberty:

> "[O]ne that breaks up from an old world, substitutes tribal support for personal merit, primogeniture right for absolute right, conformist exaltation of group values for the affirmation of self in a strange environment."

The four realities of the "free person" present before us a person who is in alliance with God, and his/her confidence in that alliance drives him/her to action. He/she is a person with free choice while those choices become part of a bigger destiny led by God. Lastly, by overcoming obstacles, this "free person" lives a new liberty, which marks a break from an old society and its values.

*Analyzing the Analysis*

Arkoun's analysis of the story of Yusuf requires its own analysis. It is an exquisite show of deconstruction. He does not follow the *sura* verse by verse or even relate the story stage after stage; rather, he finds the themes, such as freedom and the alliance between God and man, and deconstructs verses and expressions. At the same time, part of his fusion of horizons and his whole analysis revolves around the existence and presence of the divine. God is defined in his analysis as "a unique, living God who acts effectively in history" (Arkoun 2016b, p. 321). God is present in every reality of the "free person". It is the Word of God, His transcendent presence, that informs the acts of Yusuf as a role model for all men.

Arkoun goes "back and forth" between finding meaning in deconstructing details in specific words and looking at the holistic meaning. For example, in his analysis of *ghafil*, Arkoun starts from one word and quickly goes to a whole "ontologically different *after*" the Revelation, which leads to looking at the whole picture of "a struggle" between humans before and after the Revelation (Arkoun 2016b, p. 320, italics in the original text).

Arkoun's attempt to historically analyze the Founding Experience of Islam through the Prophetic Discourse is evident in his analysis of *surat Yusuf* in the next passage:

> "The narrativized symbolic values are not given to contemplate, or to an aesthetic tasting as a literary narrative; rather they are an adequate expression penetrating in more forceful meaning the socio-cultural changes effectively introduced by Muhammad in the Arab society of Hijaz" (Arkoun 2016b, p. 322).

The themes deconstructed by Arkoun are, therefore, a way to better understand the Founding Experience of Muhammad's years in Hijaz as much as they are an existential way to understand our lives.

At all stages of his analysis, Arkoun goes to the text, the OCC. He does so while opening it to deconstruction, for example, referring to *abawayhi* as a metonym; and a "retracing backwards" through Islamic Living Tradition, for example, with his reference to the Kharijite movement while writing about *la hukma illa lillah*. By bringing the Kharijite movement to the table of interpretation, Arkoun also implements his call "to hear voices reduced to silence, heterodox voices, minority voices of the vanquished and the marginalized" (Arkoun 2012, p. 33).

In his analysis of *Surat Yusuf*, Arkoun shows us a fusion of the horizons of his hermeneutic analysis. He merges concepts and terms from the Muslim religious world of thought, faith, and Traditions with concepts from the Western academic tradition. This creates an almost seamless fusion of horizons. Thus, he takes the alliance, *'ahd* or *mithaq*, and attaches to it the concept of "reciprocity of perspectives" (Arkoun 2016b, p. 321). Here 130123 1135Arkoun uses this term, which originates from Austrian phenomenologist Alfred Schutz (Schuetz 1953), to describe man's relation with the Divine in a kind of "dialogue" whose purpose is the "liberating intention of the Writings of all the practiced writings and beliefs" (Arkoun 2016b, p. 92). It is typical of Arkoun to use a very Western and even secular term to describe the relationship between Yusuf and God.

Another example is the word *tawakkul*. Arkoun adds to his translation of "putting one's confidence-hope in God" an explanation that gives an initiative tone to it. The *tawakkul*, according to Arkoun, drives the "free person", personified in Youssef as a role model for us all, to "a confident, enlightened action". This action is "the embodiment of the teaching of God" (Arkoun 2016b, p. 321). Arkoun fuses the religious horizon of trusting in

God with freedom and enlightenment, which resonate with Western Tradition, which is in itself reconnected by him to the teachings of God.

**5. Discussion: Educational Implications for RIE**

What can be the educational implications of Mohammed Arkoun's methodology of reading the Quran for religious Islamic education? The answer to this question resides in the possibility of transforming Arkoun's methodology into a pedagogical practice. Teaching the Quran along Arkounian lines will entail reading the *mushaf* (OCC in Arkounian terms), but then moving to an archaeology of the text.

In teaching each verse, one must try to (re)interpret the text by referring to the four different faces of the Quran. Relating to the Living Tradition, Law and Orthodoxy, but also bringing interpretations that were marginalized throughout the years, a teacher will analyze the verses with the students in four different ways: (1) **Semantically and linguistically**, as a discourse given to the Prophet Muhammad and recited by him to the community of believers. (2) **Through historical analysis** of the prophetic period of Mecca and Medina and the circumstances and conditions of the revelation. (3) **Axiologically**—focusing on the values that were created in a specific historical, cultural, social, and political environment and the way they are (re)embodied in Muslim living Tradition today. This brings this education to the field of *moral education*, in the terms of Althof and Berkowitz (2006). This level will also focus on the existential and reflexive qualities that the Quran carries for the lives of Muslim students today. (4) **Metaphorically**—through remembering the Marvelous and the Divine that is the origin of these texts. In a way, that could be similar maybe to *education for religion*, a type of religious education that "seeks refined moments in which the pupils reflect on the Real as it appears in their own experiences" (Agbaria 2022, p. 10). This is also the place for comparison with other monotheistic religions and the way they transform the Word of God to Living Tradition.

Such pedagogical practice entails a movement "back and forth" between analyzing each vector of the Quranic verses separately (linguistically, historically, axiologically, and metaphorically) and seeing them as a whole expressed inseparably in each Quranic verse and its meaning to the lives of the students.

Such a pedagogy would also incorporate a fusion of horizons. Combining both academic and scientific rigor with Living Muslim Tradition. This is not a teaching of the Quran from the outside or from an alienated point of view. In other words, this is not teaching *about religion.* At the same time, this is not traditional teaching and memorizing of the Quran or teaching *into religion*. Arkoun's methodology underlines the significance of critical analysis as much as it emphasizes the importance of Tradition. It also holds a significant existential and reflexive meaning for those who might use it in reading the Quran. Thus, it is best described as teaching *from religion* (Grimmit 2000).

Mohammed Arkoun's hermeneutical methodology can also have implications for what can be seen as worthy goals of RIE. The extended use of "freedom" in Arkoun's analysis is not coincidental and is representative of the period and the global space in which he lived and worked. There are many ways to interpret the story of Yusuf, and in Judaism, Christianity, and Islam, it is one of the stories that has been the subject of many commentaries and political, cultural, and artistic interpretations (Carmichael 2017). Arkoun's choice to focus on the "free person" and the realities of said freedom is not surprising from a person who lived in a country that carries that word on its banner, metaphorically and literally, and in a space that is defined by the Latin adjective derivation of the word (Liberal).

Furthermore, in Arkoun's description of Yusuf as a hero of new freedom based on personal merit rather than tribal support and an affirmation of self in a foreign environment rather than a conformist exaltation of group values, one could easily find not only the biography of Mohammed Arkoun himself, as a Muslim immigrant in Europe, but also the biographies of many Muslim immigrants to Western Europe and North America. Arkoun sees this immigration as a "unique opportunity offered to so many frustrated people,

dramatically uprooted from their own culture, land, region, and nation, to reconquer new horizons of meaning and hope for their existence" (Arkoun 2012, p. 424). Arkoun's hermeneutical analysis of the story of Yusuf could help to religiously legitimize and justify such immigration stories and give them even a sense of heroism that finds its origins in the most sacred texts and the most venerable religious figures.

Yusuf, who is a figure of a "free person" and who is representative of the Prophet Muhammad, becomes, with Muhammad, a role model, not necessarily for immediate imitation but rather in a way that "will constantly re-actualize for those who memorize it" (Arkoun 2016b, p. 322). These figures, therefore, carry a deep, complicated, and existentially reflexive meaning to the lives of Muslims throughout the generations, including our own.

The Arkounian ideal of a "free person" throughout Arkoun's work is always attached to a community. Such is his description of the community of believers who were addressed by the Prophet Muhammad as "equal and free" (Arkoun 2012, p. 70). Such is also his analysis of the term *mukallaf*. Arkoun analyzes this adjective as describing a man as "responsible in front of God" (Arkoun 2016b, p. 40), and as "the subject of rights, a person who enjoys fully the faculty of discernment (*'aql*)" (Arkoun 2016b, p. 296). He also claims that "*taklif* is a great force of social, political and religious integration" (Arkoun 2016b, p. 297). In Arkoun's *mukallaf*, we find the implementation of the autonomy that comes from a community, and so "the dynamics of belonging to communities is part of the process of individualization" (Panjwani and Agbaria 2019, p. 16).

Arkoun's educational ideal is of a person who is not free despite his community, but because of it, and in fact, he cannot be free without it. It can, therefore, be seen as an educational implementation of what Vryhof (2012) named the "rooted intellectual" or as affirming Alexander's claim (Alexander 2012) that logic is understood through history, culture, and language and comes from a certain perception of what is good. Implementing the methodology of Arkoun in reading the Quran will help create independent learners who carry their own religious knowledge and will involve students in the "why" of Islamic education, finally creating an Islamic perception compatible with democratic, pluralistic, and open societies (Saada 2018). While not ignoring the Divinity of the Quranic message and emphasizing critical thinking, Arkoun's hermeneutic methodology of reading the Quran could allow RIE to give students an experience of human freedom.

And so, we find that one of the implications that the Arkounian reading of the Quran might have on RIE is the set of goals that are Western and liberal, as much as they are part of the Muslim religious Tradition. Muslim students should aspire to initiative because of *tawakkul*, and they should be "free" because of their community and religious beliefs. Arkoun sketches for us the images that are relevant to these ideals in the Quranic stories and urges us to be reflexive in the way we read these stories.

## 6. This Is Not a Conclusion (Nor a Pipe)

Mohammad Arkoun presents a hermeneutical methodology of reading the Quran in a way that can enable teachers and students of RIE to teach and learn in a creative and meaningful way. Analyzing critically both the Prophetic Discourse and the Islamic Tradition is a methodology whose tools of deconstruction and critical analysis are embedded in academic research and the scientific world. At the same time, working through and with the Muslim religious Tradition and trying to find meaning in the transcendent Word of God and our relationship with it is a methodology that is not only respectful of faith and tradition but also embedded in Islamic thought and Tradition.

Arkoun also articulates goals for such a reading of the Quran as well as a religious education. Creating free-thinking students who are free in their thought because they are a part of a community, whose autonomy and authenticity are rooted in a sense of belonging, and who are aware of the wonder and divinity in their lives and the endless ways to interpret that wonder.

Implementing Arkoun's philosophy in educational settings does entail, in our opinion, two more difficulties that should be addressed. The first is that education cannot include

everything. Like the move from oral to written culture described by Arkoun, teaching means choice and is, therefore, submitted to the dialectics of power and residue. Crystalizing a pedagogy along Arkounian lines would necessarily mean the crystalizing of an *unthought*. In doing so, we should always be aware of that and look to expand what we teach and learn and how we teach and learn it.

The second difficulty is the "glass ceiling" of Arkoun's criticism. There are ontological assumptions that Arkoun does not expose to deconstruction and critical analysis. No criticism or even question is proposed as to the existence of God, nor is the Quran as part of His message. One of us (Felsenthal) finds this to be lacking and incoherent with Arkoun's message of deconstruction and criticism. The other (Agbaria) finds this to be a strong point in the possibility of implementing Arkoun's methodology in RIE in general and in confessional education. It may be best to view this as a tension. Arkoun's hermeneutical methodology of reading the Quran is a mélange of not just Islamic and Western thought but of secular and religious points of view. It is this kind of mélange that can make the tension between the worldviews fruitful. It remains, therefore, an open and fertile question as to effect how the Quran is read, taught, and learned in RIE.

One can already imagine a classroom teaching the Qur'an according to Arkounian methodology. A rigorous analysis of words and verses from the Qur'an, discussions on the historical circumstances in seventh-century Hijaz, a moral debate about values, and also a sharing of Living Traditions. One can see a teacher helping students to become free critical subjects who are also part of a community.

**Author Contributions:** Conceptualization: I.F. and A.A.; methodology I.F. and A.A.; investigation, I.F. and A.A.; resources, I.F. and A.A.; writing—original draft preparation, I.F. and A.A.; writing—review and editing, I.F. and A.A.; supervision, A.A.; project administration, I.F.; funding acquisition, I.F. and A.A. All authors have read and agreed to the published version of the manuscript.

**Funding:** This research was funded by the Gerda Henkel Stiftung, grant number AZ 35/V/22.

**Conflicts of Interest:** The authors declare no conflict of interest.

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
