# Peer review of "How to Read the Quran in Religious Islamic Education: What Educators Can Learn from the Work of Mohammed Arkoun"

_religions, doi:10.3390/rel14010129_

Round 1

Reviewer 1 Report

The author sets out to explain how Mohammed Arkoun's framework for Quranic interpretation can be useful in Islamic education. The author generally does a good job of explaining and demonstrating Arkoun's approach. They also make clear what Arkoun is trying to achieve and how his approach differs from other approaches.

If I see a shortcoming in the paper, it is that most of it is dedicated to explaining Arkoun's interpretive approach and philosophy, while the question of whether and how Arkoun's approach can contribute to Islamic education is addressed only briefly at the beginning and the end of the paper. The paper would be stronger if the authors' arguments for why and how Arkoun can transform the way that interpretation is taught in Islamic educational institutions were threaded throughout the paper rather than confined to the introduction and the conclusion. 

Author Response

Thank you for your review. We added in some sections sentences concerning how Arkoun's analysis can be used in RIE. 

Reviewer 2 Report

This article constitutes an interesting contribution to understanding Mohammed Arkoun's philosophical thoughts connected with Islamic pedagogical practices. Some minor issues should be taken into consideration before publication.

In p. 6: the author wrote: "At this level, the Quran which up to this stage was a verbal discourse, is written down and becomes mushaf. This is, in fact, the written text of the Qur'an which we have before us today." The reviewer is uncertain that Arkoun really said this, or this is the author's interpretation of what Arkoun meant with a closed corpus. In fact, In the Muslim view, the recited Qur'an matters. Moreover, Muslims felt more comfortable with oral transmission of the Qur'an along with the standarisation or canonisation projects that happened during the first three centuries of Islamic periods, and then in the 1920s, the Qur'an received its final standarisation using one reading system only. 

In p. 7: the author wrote: "It is the third Caliph, Othman, who collected and edited the Qur'an." This sentence is incorrect and leads to misunderstanding about the history of the Qur'an in early Islamic period. Please, be more precise in describing the canonisation of the Qur'an, especially that was initiated by the Caliph Uthman. The author is advised to refer to some relevant scholarly publications. One example is Rezvan's work, "The Qur'an and its world: VI. Emergence of the canon: the struggle for uniformity."

There are some typographical errors. First, in p. 2, it is written: "Thilal al-Qur'an by Sayyid Qutub." Does the author mean "Fi Zilal al-Qur'an"? Second, in p. 5, the author wrote: "... the societies of the Book-book..." Is this "...the societies of the Book..."? Some other errors are found in several pages that need correction or careful copy-editing.

Author Response

Thank you for your review. 

1. Arkoun refers to the mushaf as a written corpus and we have added a reference from his work. 
2. The emphasis that Arkoun makes is to the "imperial moment" and the way it affected the creation of the OCC, and not to the third Caliph. You are right and we have removed that sentence. 
3. We have corrected the title by Qutb to Fi Zilal al-Quran. 
4. We have also corrected other typographical errors

Reviewer 3 Report

This paper is a very important contribution to the field. Pedagogy in Islamic education and Qur'anic studies are critical areas of research and there is an urgent need for innovative thinking. Although I don't necessarily agree with Arkoun's dismissal of the Qur'anic exegetic tradition, I do feel this paper provides a much needed perspective to the debate. The paper is very well structured, Arkoun's key ideas related to Quran'ic interpretation are cogently outlined, and both context (opening) and way forward (closing sections) provide serious food for thought - particularly for educators in schools who can be using an article like this for professional learning. A strong contribution from start to finish.  

Author Response

Thank you for your review. Indeed, we hope that this approach will be useful for educators.